# Leverage of Environmental Pollutant Crump Rubber on the Dry Sliding Wear Response of Epoxy Composites

**DOI:** 10.3390/polym13172894

**Published:** 2021-08-27

**Authors:** Kiran Shahapurkar, Venkatesh Chenrayan, Manzoore Elahi M. Soudagar, Irfan Anjum Badruddin, Pavan Shahapurkar, Ashraf Elfasakhany, MA Mujtaba, Md Irfanul Haque Siddiqui, Masood Ashraf Ali, Teuku Meurah Indra Mahlia

**Affiliations:** 1School of Mechanical, Chemical and Materials Engineering, Adama Science and Technology University, Adama 1888, Ethiopia; venkymech0607@gmail.com; 2Department of Mechanical Engineering, School of Technology, Glocal University, Delhi-Yamunotri Marg, SH-57, Mirzapur Pole, Saharanpur District, Uttar Pradesh 247121, India; me.soudagar@gmail.com; 3Mechanical Engineering Department, College of Engineering, King Khalid University, Abha 61421, Saudi Arabia; magami.irfan@gmail.com; 4Applied and Theoretical Mechanics Laboratory, Discipline of Mechanical Engineering, Indian Institute of Technology, Indore 453552, India; shahapurkarpa1@gmail.com; 5Mechanical Engineering Department, College of Engineering, Taif University, P.O. Box 11099, Taif 21944, Saudi Arabia; a.taha@tu.edu.sa; 6Department of Mechanical Engineering, Faculty of Engineering, University of Malaya, Kuala Lumpur 50603, Malaysia; m.mujtaba@uet.edu.pk; 7Department of Mechanical Engineering, College of Engineering, King Saud University, Riyadh PIN 11451, Saudi Arabia; msiddiqui2.c@ksu.edu.sa; 8Department of Mechanical Engineering, College of Engineering, Prince Sattam Bin Abdulaziz University, Al Kharj 16273, Saudi Arabia; mas.ali@psau.edu.sa; 9Centre for Green Technology, Faculty of Engineering and Information Technology, University of Technology Sydney, Sydney, NSW 2007, Australia

**Keywords:** environmental pollutant, polymer matrix composites, crump rubber, wear

## Abstract

The effect of crump rubber on the dry sliding wear behavior of epoxy composites is investigated in the present study. Wear tests are carried out for three levels of crump rubber (10, 20, and 30 vol.%), normal applied load (30, 40, and 50 N), and sliding distance (1, 3, and 5 km). The wear behavior of crump rubber–epoxy composites is investigated against EN31 steel discs. The hybrid mathematical approach of Taguchi-coupled Grey Relational Analysis (GRA)—Principal Component Analysis (PCA) is used to examine the influence of crump rubber on the tribological response of composites. Mathematical and experimental results reveal that increasing crump rubber content reduces the wear rate of composites. Composites also show a significant decrease in specific wear values at higher applied loads. Furthermore, the coefficient of friction also shows a decreasing trend with an increase in crump rubber content, indicating the effectiveness of reinforcing crump rubber in a widely used epoxy matrix. Analysis of Variance (ANOVA) results also reveal that the crump rubber content in the composite is a significant parameter to influence the wear characteristic. The post-test temperature of discs increases with an increase in the applied load, while decreasing with an increase in filler loading. Worn surfaces are analyzed using scanning electron microscopy to understand structure–property correlations. Finally, existing studies available in the literature are compared with the wear data of the present study in the form of a property map.

## 1. Introduction

Polymeric composite materials have gained significant importance in contemporary manufacturing scenarios, due to a number of advantages offered in terms of low density, high strength-to-weight ratio, and higher specific properties [1]. Composite materials are prepared by reinforcing particles in a resinous matrix [2]; these composites are employed in numerous marine, transportation, and aerospace operations [3]. Polymer composites are increasingly being used as friction materials in brake liner applications [4]. Investigations into the wear response of polymer composites as tribomaterials are increasing compared to metal matrix composites, due to their growing demand in the transportation and aviation sectors [5]. Wear is one of the most common occurrences in tribology-related applications and, therefore, an ever-increasing push is seen towards developing materials with low coefficient of friction and cost of constituents, ease of manufacturing, and higher specific properties [6]. Of late, significant research has been carried out with waste materials to enhance the properties of materials. These waste materials, derived from either industrial or natural waste, create a lot of landfill and disposal issues. Thus, much research is dedicated to overcoming these shortcomings by developing utilitarian composites. Industrial waste products such as blast furnace slag [7], Linz–Donawitz slag [8], cenospheres [9,10,11,12,13], red mud [14], and fly ash [15] are effectively utilized to produce composites with improved mechanical characteristics [16,17]. Furthermore, the low cost of fillers is generally attributed to their high availability. One such low-cost, environmental contaminant filler is crump rubber. Waste tires are a significant source of crump rubber. The landfill burden and disposal issue of waste tires has emerged as a severe problem across the globe. The abundant waste tires, which possess a very long life, are non-biodegradable and toxic, requiring effective disposal solutions [18]. Crump rubber is derived from grinding of waste tires. Crump rubber is composed of carbon and oxygen as its main constituents (74%), whereas aluminum, silicon, sulfur, calcium, titanium, iron, and zinc are present in trace quantities [19]. Based on the available studies in the literature, it can be inferred that a combination of experimental and mathematical modelling studies on the dry sliding wear response of crump rubber–epoxy composites has not been reported to date; hence, the present study intends to fill this gap.

The present study aims to understand the effect of crump rubber on the wear behavior of particulate-filled polymer composites and demonstrate the important parameters that help to develop good friction materials. In this regard, a pin-on-disc setup was utilized to test the composites under dry conditions. Although a pin-on-disc setup does not reproduce every aspect of an envisaged application, pin-on-disc tests are widely utilized to examine the wear response of materials designed for braking systems in many transportation sectors, such as trains and automobiles under steady-state surroundings. Additionally, pin-on-disc tests help to correlate the structure–property relations of materials, and are useful in modelling wear and lubricant response in braking pads with linear relative velocity [20].

The development of crump rubber/epoxy composites serves the twin purpose of effectively utilizing environmental pollutant crump rubber and lowering component costs. In the present investigation, the leverage of crump rubber content on applied load and sliding distance on the rate of wear and frictional coefficient are extensively studied with the help of the Taguchi-coupled GRA-PCA technique. Furthermore, the morphology of worn-out surfaces and post-wear debris is analyzed using a scanning electron microscope. Finally, results from the current study are compared with the available studies, and are depicted as a property map to act as a guide for industrial practitioners and researchers.

## 2. Materials and Methods

### 2.1. Materials

In the present investigation, LAPOX L-12 epoxy resin with K6 hardener was used as a matrix, procured from Atul Industries Ltd., Valsad, Gujarat, India. Crump rubber particles of 180 μm size were used as reinforcement, supplied by Arihant Chemicals Ltd., Delhi, India. Particle size analysis and a micrograph of crump rubber are shown in Figure 1. Properties of crump rubber are depicted in Table 1.

### 2.2. Composite Fabrication

Composites were manufactured with varying crump rubber contents of 10, 20, and 30 vol.%. Crump rubber of 10, 20, and 30 vol.% was dispersed in the epoxy matrix to acquire homogeneous and consistent slurry. Then, 10 wt.% of K6 hardener was added to the slurry and de-aerated for 5 min prior to pouring into aluminum molds with dimensions of 90 × 90 × 12 mm, coated with a silicone-releasing agent. Cast slabs were cured for 24 h under ambient conditions, and samples were trimmed to a uniform size as per the ASTM G99-17 standard (12 × 12 × 25.4 mm thick). Coding of samples was carried out as per EC-XX convention, where E, C, and XX represent epoxy, crump rubber, and filler content by vol.%, respectively.

### 2.3. Wear Test

Wear tests were performed in ambient settings using a pin-on-disc tribometer procured from DUCOM, Bengaluru, India (Figure 2). To test the developed samples, and EN-31 disc with 62 HRC hardness and 0.11 µm roughness was used, as rotor discs in automotive brake systems are mostly prepared from grey cast iron and steel, owing to their exceptional damping capabilities and high resistance to wear and tear [21,22]. Tests were performed on a track with a diameter of 120 mm at 795 rpm, corresponding to a 5 m/s sliding velocity. Table 2 depicts the levels of parameters used for testing. Sliding velocity, sliding distance, and applied normal load are shown as Vx, Dy, and Fz, respectively, where x, y, and z represent the values of the parameters. In the present study, three average nominal contact pressures Pn—i.e., 0.21, 0.28, and 0.35 MPa—were studied equivalent to normal loads of 30, 40, and 50 N, respectively. These loads were selected to replicate low and moderate sliding wear circumstances for braking applications, wherein typical values of the product of contact pressure and sliding velocity vary from 0.3 to 20 MPa-m/s. Before each test, the disc was polished with emery paper and cleaned with acetone to maintain an even surface roughness value of 0.11 µm. Then, the sample was firmly clamped in the holder, and the tests were performed as described in the ASTM G99-17 standard. Three samples were tested for each configuration, and the average values are presented here.

Height loss and frictional forces of the tests were recorded using a data acquisition system connected to the wear setup. The cross-sectional area of the pin was used to calculate the volume loss.

Wear rate ($W_{r}$) is given as:(1)$$W_{r}=\frac{V_{e}-V_{s}}{S_{e}-S_{s}}$$
where V and S correspond to volume loss and sliding distance, respectively, whereas s and e correspond to the end and start of the steady-state wear, respectively.

The load-carrying capacity of samples is determined by the specific wear rate (W_s_), and is given as:(2)$$W_{s}=\frac{\mathrm{Wr}}{F}\;$$

The coefficient of friction (COF) (μ) is given as:(3)$$\mu =\frac{F_{t}}{F_{n}}$$
where $F_{t}$ and $F_{n}$ are tangential and normal force, respectively.

### 2.4. Temperature Measurement and Scanning Electron Microscope

The temperature of the EN-31 steel disc after each test was measured using a thermocouple (CMPH–x, HAMI THERM, Dordrecht, Netherlands). Surfaces of specimens subjected to wear were analyzed using a scanning electron microscope (JSM 6380LA, Jeol Ltd., Tokyo, Japan). Specimens were sputter-coated to give them a conducting layer (JFC-1600, Jeol Ltd., Tokyo, Japan).

## 3. Statistical Evaluation

### 3.1. Taguchi-Coupled GRA Principal Component Analysis

#### 3.1.1. Taguchi’s Design of Experiment

It is essential to address the combination of experiments by considering the number of dependent and independent variables at their levels. The number of experiments can be minimized with the implementation of this technique [23,24]. The L9 orthogonal array was followed for the present work with three parameters and two objectives. Table 3 shows the noted values of the two objectives—namely, the specific wear rate and coefficient of friction for each combination of wear test.

#### 3.1.2. Grey Relational Analysis (GRA)

GRA is a useful technique that was advocated by Professor Deng (1989) [25] to deal with complex problems with a limited number of available data. The concept of GRA theory starts with normalizing the raw data. Another critical aspect of GRA is the capability to solve multi-objective cases by forming them into a single objective [26,27,28,29]. The signal-to-noise ratio is manipulated by following the recommended equation according to the expectation of the performance of responses, whether smaller or larger [30,31,32]. The present study involves the tribological behavior, which requires both responses to be as low as possible. The S/N ratio for each experimental run is calculated as:(4)$$\frac{S}{N}\;\mathrm{ratio}=-10\log\frac{1}{n}\;\overset{n}{\underset{i=1}{\sum }}Y_{i}^{2}$$

The data are normalized from 0 to 1 by utilizing the following equation:(5)$$Z_{ij}=\frac{\max\left(Y_{ij},i=1,2,\ldots ,n\right)-Y_{ij}}{\max\left(Y_{ij},i=1,2,\ldots ,n\right)-\min\left(Y_{ij},i=1,2,\ldots ,n\right)}$$

The grey relational coefficients are determined with the following equation by incorporating the appropriate quality loss ($\Delta_{\min,\;}\;\Delta_{\max}$).
(6)$$GC_{ij}=\frac{\Delta_{\min}+\lambda \Delta_{\max}}{\Delta_{ij}+\lambda \Delta_{\max}}$$

Table 4 shows the computed grey coefficient value, S/N ratio, and normalized S/N ratio.

#### 3.1.3. Principal Component Analysis (PCA)

PCA is one of the most important mathematical techniques to minimize the dimensionality of data collection by retaining the deviations that exist in the data pool [33,34,35]. The conversion of recorded responses through this mathematical approach paves the way to generate a new data group, called principal components. The normalized data of recorded responses are utilized to compute the value of GRG. The GRG is used to construct the covariance matrix as per the PCA model. The eigenvectors of the transformed matrix are manipulated and the corresponding eigenvalues are also computed with the help of Equation (7), shown below [36]:(7)$$\left|\begin{array}{ccccc} x_{1}\left(1\right) & x_{1}\left(2\right) & - & - & x_{1}\left(n\right)\\ x_{2}\left(1\right) & x_{2}\left(2\right) & - & - & x_{2}\left(n\right)\\ - & - & - & - & -\\ - & - & - & - & -\\ x_{m}\left(1\right) & x_{m}\left(2\right) & - & - & x_{m}\left(n\right) \end{array}\right|$$
where, $x_{i}\left(j\right)$ represents the value in the matrix, *i* = 1,2,3,…*m*, and *j* = 1,2,3,…*n.*

X is the value of each grey relational coefficient corresponding to each response, “*m*” is the number of experiments, and “*n*” is the number of responses.

The coefficient of the correlation array can be determined as:(8)$$R_{Jl}=\frac{Cov\left(x_{i}\left(j\right),x_{i}\left(l\right)\right)}{\sigma x_{i}\left(j\right)\times \sigma x_{i}\left(l\right)}\;\;\;$$
where $Cov\left(x_{i}\left(j\right),x_{i}\left(l\right)\right)$ is the covariance sequence of *x_i_*(*j*) and *x_i_*(*l*), and σ *x_i_*(*j*), σ *x_i_*(*l*) are the standard deviations of the sequence *x_i_*(*j*) and *x_i_*(*l*)*,* respectively.

Then, the eigenvectors and eigenvalues are determined by using a coefficient correlation array with the help of the equation shown below:(9)$$\left(R-\lambda_{k}I_{m}\right)V_{ik}=0$$
where $\lambda_{k}$ expresses the eigenvalues—$\sum_{k=1}^{n}\lambda_{k}=n$; *k* = 1,2,3…*n*—and the entity $V_{ik}$ is the eigenvector $V_{ik}$ = ${\left[a_{k1,\;}a_{k2,\ldots \ldots \ldots }a_{kn}\right]}^{T}$ for the corresponding eigenvalues of $\lambda_{k}$.

The principal component of each response value can be determined as follows:(10)$$\mathrm{Principal}\;\mathrm{component}\;Y_{mk}=\sum_{i=1}^{n}X_{m}\left(i\right)V_{ik}$$

The principal component value of each response for an individual experimental run is shown in Table 5, and the eigenvalue of each principal component is shown in Table 6.

#### 3.1.4. Manipulation of Grey Relational Grades (GRGs)

After arriving at the weightage value of the principal component of each response as shown in Table 7, the grey relational grade can be determined for each response using the following equation [37]:(11)$$\gamma_{i}=\frac{1}{n}\;\sum_{j=1}^{n}\beta_{k}\left(GC_{ij}\right)$$
where $\gamma_{i}$ is the grey relational grade of the *i*th experiment, *n* is the number of responses, $\beta_{k}$ is the weightage of the principal component, and $GC_{ij}$ is the grey relational coefficient of the *j*th response in the *i*th experiment. The computed GRGs and their rankings are plotted in Table 8.

#### 3.1.5. Optimal Combination of Input Parameters and Their Levels

The optimal combination of wear test parameters and the percentage composition of crump rubber to yield a low wear loss and friction coefficient can be estimated through GRG analysis. The grey relational grade acts as an index, which is arrived through grey relational coefficient and principal component weightage, used to imply the degree of quality characteristics. A ranking was assigned to each experimental run subject to the GRG values.

Table 8 shows that experimental number 8 has the highest GRG ranking which, in turn, expresses a better-quality characteristic. The combinations of experiment 8 were 30 vol.% of crump rubber, 3 km of sliding distance, and 50 N of normal load, which are considered to be optimal wear test parameters. The main effects plot plotted for the data means and GRGs is shown in Figure 3. The main effects plot acknowledges the optimal input parameters.

#### 3.1.6. ANOVA Analysis to Study the Effect of Parameters over the Response

Analysis of variance is carried out with the first principal component, which has the most prominent weightage. The principal component with the largest eigenvalues always retains the highest variance [38,39].

Table 9 and Table 10 show the ANOVA analysis for the specific wear rate and coefficient of friction, respectively.

The ANOVA analysis explores the role of parameters on the effects of performance characteristics. The R^2^ and R^2^ (adj) values arrived at for the analysis were nearly 90%, which indicates that the proposed mathematical model is good [40]. The research work upholds the importance of the inclusion of crump rubber to minimize the wear rate. ANOVA analysis shows that the increased composition of crump rubber in the developed epoxy matrix plays a more instrumental role in improving the tribological properties than any of the other parameters. The majority of the contribution to influence both specific wear rate and coefficient of friction is from the crump rubber composition. The wear test parameters sliding distance and normal load were the least significant in this study. Figure 4 depicts outlier and normal probability plots where the points are linearly arranged, indicating the adequacy of the proposed model with a 95% confidence interval.

The developed mathematical employed the following regression equation to study the relationship between wear test parameters and responses:Specific wear rate = 0.5456 − 0.1479 x1 − 0.0125 x2 + 0.1604 x3 − 0.0261 y1 + 0.0684 y2 − 0.0424 y3 − 0.0835 z1 − 0.0932 z2 + 0.1767 z3(12)
Coefficient of friction = 0.4818 − 0.1453 x1 − 0.0507 x2 + 0.196 x3 + 0.066 y1 − 0.039 y2 − 0.027 y3 + 0.056 z1 − 0.024 z2 − 0.032 z3(13)
where x1, x2, and x3 are the percentage of crump rubber at levels 1, 2, and 3, respectively.where y1, y2, and y3 are the sliding distances at levels 1, 2, and 3, respectively.where z1, z2, and z3 are the normal load at levels 1, 2, and 3, respectively.

## 4. Results and Discussion

### 4.1. Effect of Parameters on the Responses

#### 4.1.1. Wear Rate

Tests were performed at a constant sliding velocity because load and sliding distance significantly affect the wear of samples compared with other parameters. The wear rates of crump-rubber-reinforced epoxy samples are depicted in Figure 5.

The wear rate of samples reveals decreasing trends with the increase in crump rubber content and applied normal load. The same phenomenon is acknowledged by the statistical analysis. ANOVA results show that the crump rubber content has the maximum contribution (52.60%) to influence the wear rate. Similarly, the normal load has a secondary contribution (42.00%) over the wear rate. However, the effect of sliding distance seems to be the least significant (5.40%) in the mathematical model. Maximum wear of 7.6 mm^3^/km was observed in the V5D5F30 condition for EC-10, whereas EC-30 showed minimum wear of 0.9 mm^3^/km in the V5D1F50 condition. Epoxy is brittle by nature and reveals a higher wear rate, as reported in [41]. Therefore, reinforcement with crump rubber particles in brittle epoxy leads to a lower wear rate. Increasing the crump rubber content (up to 30 vol.%) leads to better compliance of the composites to wear, and further enhancement in wear resistance. Wear rate reduces with an increase in load from 30 to 50 N. Higher loads are known to induce more volume loss in epoxy composites, but reinforcement with crump rubber particles reduces the surface asperities by absorbing the load effectively, thereby assisting in lowering the wear rate. Increasing the crump rubber content further reduces the wear rate. Another reason for the decrease in wear rate is the decrease in brittle material (epoxy) with increased ductile material (crump rubber). Therefore, observed results show a significant reduction at EC-30. Comparing the wear rates of EC-10, EC-20, and EC-30 reveals that by increasing the load from 30 to 50 N, the wear rate is reduced in the range of 90–122%, while increasing the crump rubber content from 10 to 30 vol.% reduces the wear rate in the range of 100–122%. This experimental effect coincides with the results of mathematical modeling. These observations indicate the effectiveness of reinforcement with crump rubber in the most widely used epoxy matrices, making them potential materials for dry sliding wear conditions.

#### 4.1.2. Specific Wear Rate

The load-bearing capacity of the samples is measured by the specific wear rate. The specific wear rate of the samples is presented in Figure 6.

A decrease in the specific wear rate of all of the samples is seen with the increase in load. A substantial decrease is observed at EC-30, showing higher wear resistance at higher loads. A minimum specific wear rate of 0.02 mm^3^/N-km is observed with EC-30 in the V5D1F50 condition.

#### 4.1.3. Coefficient of Friction (μ)

The μ observed for different loading conditions is shown in Figure 7.It can be seen that μ shows a decreasing trend with an increase in crump rubber content. Higher filler loadings result in minimal variations at the interface between the pin (sample) and the rotary disc. Thus, lower surface roughness values are seen and, consequently, μ decreases. μ varies from 0.12 to 0.39 for all of the test conditions. Furthermore, it should be noted that μ values show significant reduction at higher applied loads (50 N), as supported by the mathematical modelling results. The optimal parameters recorded for a minimum level of coefficient of friction in PCA analysis are a higher level of crump rubber content (30%), medium level of sliding distance (3 m/s), and a higher level of normal load (50 N). An increase in contact between the sample and the disc leads to effective compression of asperities on the disc, and results in the formation of a lubricating film. Therefore, μ is considerably reduced at higher loads with more crump rubber content, and the same is reflected in the wear of the samples.

#### 4.1.4. Disc Temperature

The temperature of the disc recorded after each test for different crump rubber loadings and test conditions is reported in Figure 8. For all of the specimens, disc temperature increased with an increase in applied load, attributed to higher frictional forces and high adiabatic heating at the interface of the disc and the specimen. The increase in disc temperature is a strong function of applied load and sliding velocity. Nevertheless, it can be seen that disc temperatures recorded with increasing filler loadings show decreasing trends. These trends are attributed to the ability of crump rubber particles to effectively absorb the heat during sliding wear. Thermogravimetric analysis curves reported in [42] reveal that all crump rubber–epoxy composites reveal a one-step degradation mechanism, and onset of degradation for EC-10, EC-20, and EC-30 takes place at 359, 362, and 365 °C, respectively. Furthermore, we also found that EC-30 registers a higher final degradation temperature, and char residue left at 700 °C is also higher compared to lower filler loadings, indicating higher thermal stability. Furthermore, we also predict that low undulations of the pin on the disc with higher filler content will restrict the rise in temperature. Although temperature plays a significant role in dry sliding wear scenarios, the present study demonstrates that reinforcement with crump rubber restricts the rise in temperature to a significant degree. Differential scanning calorimetry tests confirm that the glass transition temperature of EC-10, EC-20, and EC-30 composites is 85, 89, and 101–105 °C, respectively. Furthermore, wider curing curves are seen with crump rubber composites, depicting complete polymerization of composites by releasing more heat during curing. Therefore, it can be concluded that crump rubber–epoxy composites can be considered to be best suited for dry sliding wear scenarios, given their excellent ability to resist sliding wear, even under elevated-temperature environments, owing to their high thermal stability.

### 4.2. Wear Debris

Analysis of wear debris found on the disc surface after each test can lead to better understanding of the wear mechanisms. The morphology of wear debris as seen from the scanning electron microscope is depicted in Figure 9. Wear debris of EC-10 at 30 N is shown in Figure 9a; the size of the debris, as evident from the micrograph, is larger, attributed to higher epoxy content and dominance of the brittle response to dry sliding wear. Contrastingly, at 50 N applied loading, the debris is observed to be smaller in size, and the appearance of the debris is more circular in nature (Figure 9b), compared to 30 N applied loading (Figure 9a). These variations in the size of debris are attributed to higher applied loads, as the surface asperities are effectively compacted and, thus, lower wear rate is observed at elevated loads. Similarly to EC-10, EC-30 specimens also reveal debris of similar kinds (Figure 9c,d). However, some subtle changes should be noted. The size and shape of wear debris are considerably smaller due to the higher crump rubber content constraining the size of the wear debris formed. These observations reveal the effectiveness of using crump rubber in wear scenarios, as the higher crump rubber content reduces the wear rate and provides effective assistance in absorbing the undulations of the pin on the disc.

### 4.3. Scanning Electron Micrographs

Worn-out surfaces of specimens subjected to different loads, analyzed using a scanning electron microscope, are presented in Figure 10. Micrographs of EC-10 at 30 and 50 N are depicted in Figure 10a,b, respectively. It can be seen that the wear tracks are severe in appearance and reveal higher wear loss, as evident from Figure 5. Low crump rubber content and dominance of the brittle epoxy matrix tend to increase the wear. However, it can be seen that the specimen surfaces at 50 N (Figure 10a) are less severe compared with 30 N (Figure 10b). This is attributed to the fact that 40 N applied loading is adequate to compact the surface asperities and provide a stable wear rate [6,12,43]. Micrographs of EC-20 at 30 and 50 N applied loading are shown in Figure 10c and Figure 10d, respectively. It can be seen that the content of crump rubber seems to be sufficient to have a significant effect on the wear response of the composites. At 30 N applied loading, wear debris owing to the sliding of the constituents is visible on the surface (Figure 10c). At 50 N applied loading, it can be seen that crump rubber particles, due to their inherent elastic nature, tend to form a layer on the surface of the specimen (Figure 10d). A combination of higher loadings and higher crump rubber content leads to the formation of a layer on the surface of the specimen. These observations are similar to previous investigations [5,6,44,45,46]. Micrographs of EC-30 at 30 and 50 N are depicted in Figure 10e,f, respectively. The formation of a thin film on the surface of the specimen is considered advantageous in minimizing the wear rate. At 30 N, the thin film’s formation can be observed as the higher filler content is dictated by the wear behavior (Figure 10e). These observations are consistent with the observed wear rate in Figure 5. As the crump rubber is more elastic in nature, it tends to decrease the wear rate of the composites by converting the brittle behavior of the epoxy matrix to be more ductile. Furthermore, crump rubber content is sufficient to dominate the wear behavior and form a thin layer on the specimen’s surface. At 50 N applied loading, the combination of higher loading and crump rubber compact the wear surfaces effectively and, thus, show decreased wear rate compared to other configurations. EC-30 is best suited for dry sliding wear applications. The utilization of crump rubber particles for dry sliding wear scenarios is evident from the study, and the observed results can be useful in developing effective composite materials for potential tribological applications.

## 5. Property Map

Quantifying the observed results with available studies in the form of a property map acts as a guideline for researchers and industries in selecting a specific configuration based on the envisaged applications. Wear rate results recorded at a velocity of 5 m/s and an applied load of 50 N are compared with the extracted data from the literature and are plotted against density in Figure 11 [12,47,48,49,50,51]. Data extracted from metal matrix composites are also mapped in Figure 11 to show the effectiveness of the present work. It can be seen that polymer matrix composites show lower wear rates and density compared to metal matrix composites. However, comparing the present study’s results with fly ash, rice husk ash, cenosphere-based epoxy, and vinyl ester composites shows intermediate values. Nevertheless, the advantages of reinforcement with waste and non-disposable crump rubber in dry sliding wear scenarios are clearly evident from Figure 11. EC-30 reveals the lowest wear rate from the present study and can be considered to be best suited for wear applications. Crump rubber is abundantly available throughout the world and can be effectively used to fabricate composites for wear-resistant applications and minimize the landfill burden and concerned disposal issues.

## 6. Conclusions

In the present study, the dry sliding wear behavior of crump-rubber-reinforced epoxy composites was investigated for varying applied loadings, sliding distances, and filler contents by using GRA-coupled PCA analysis. The following conclusions can be drawn:Wear rate decreases with an increase in crump rubber content from 10 to 30 vol.%; the reduction is in the range of 100–122%.The specific wear rate of samples also shows a decreasing trend in line with wear rate. In addition, a significant reduction in the coefficient of friction is also observed with higher applied loading, which is attributed to the formation of a film between the contact interfaces.EC-30 composites reveal the highest wear resistance and are well suited for dry sliding wear conditions. In addition, an increase in applied loading and filler content shows lower values of coefficient of friction attributed to the formation of a film between the interfaces.The main effects plot drawn for GRGs acknowledges that the higher content of crump rubber and a higher level of normal load contributed significantly to reducing the specific wear rate and coefficient of friction.ANOVA analysis also shows the importance of increased crump rubber content to yield an efficient wear rate and coefficient of friction.The outlier and normal probability plots confirm the satisfactory execution of the proposed model through the non-scattered distribution of points.The post-test temperature of discs reveals increasing trends with an increase in the applied load and decreasing trends with an increase in crump rubber content.

## Figures and Tables

**Figure 1 polymers-13-02894-f001:**
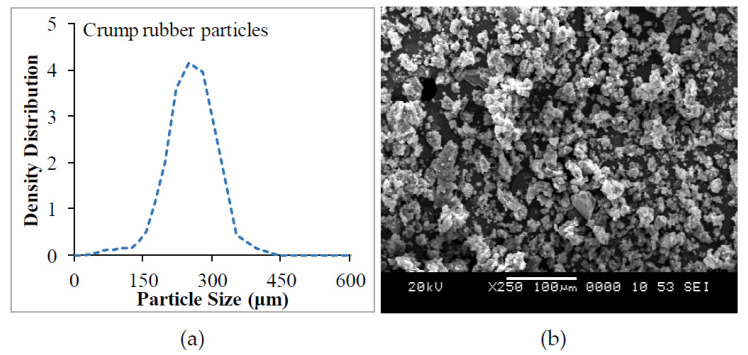
(**a**) Particle size analysis and (**b**) micrograph of crump rubber.

**Figure 2 polymers-13-02894-f002:**
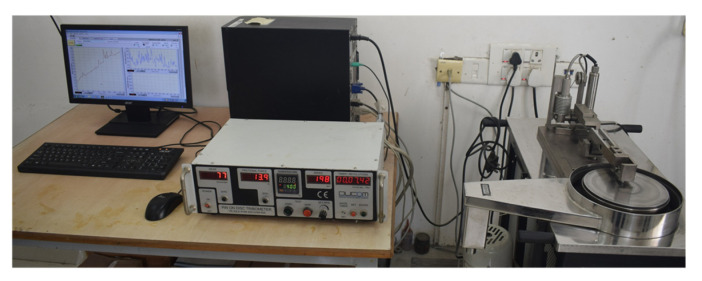
Dry sliding wear experimental setup [5].

**Figure 3 polymers-13-02894-f003:**
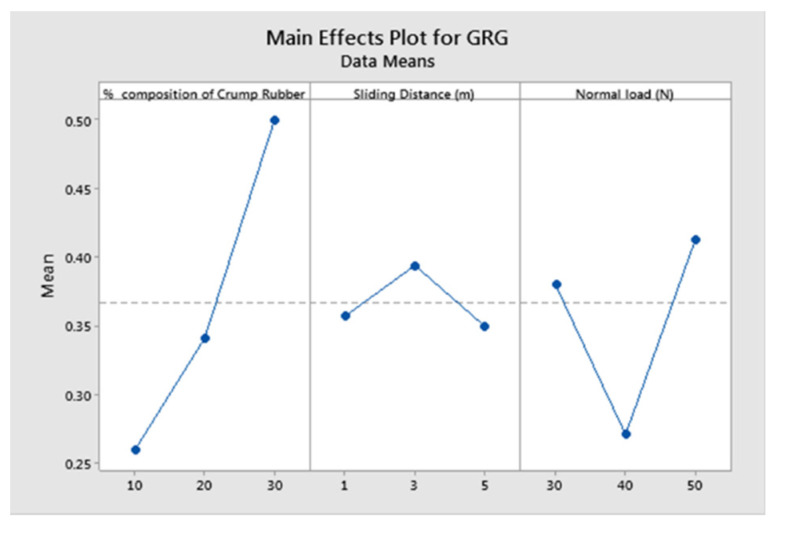
Main effects plot for GRGs.

**Figure 4 polymers-13-02894-f004:**
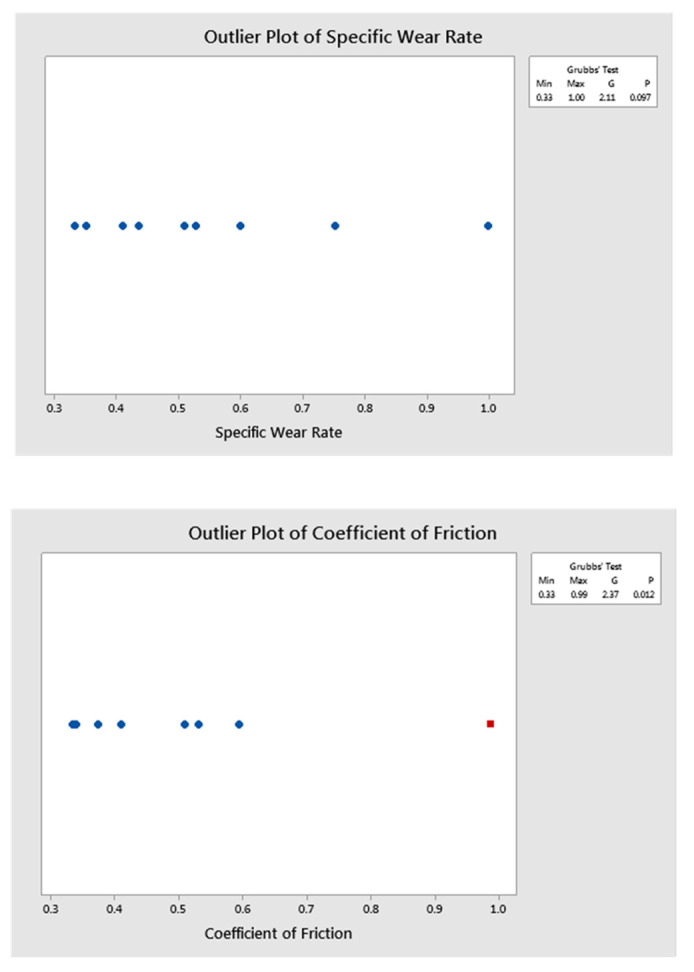
Plots of outlier and normal probability for the proposed GRA—PCA method.

**Figure 5 polymers-13-02894-f005:**
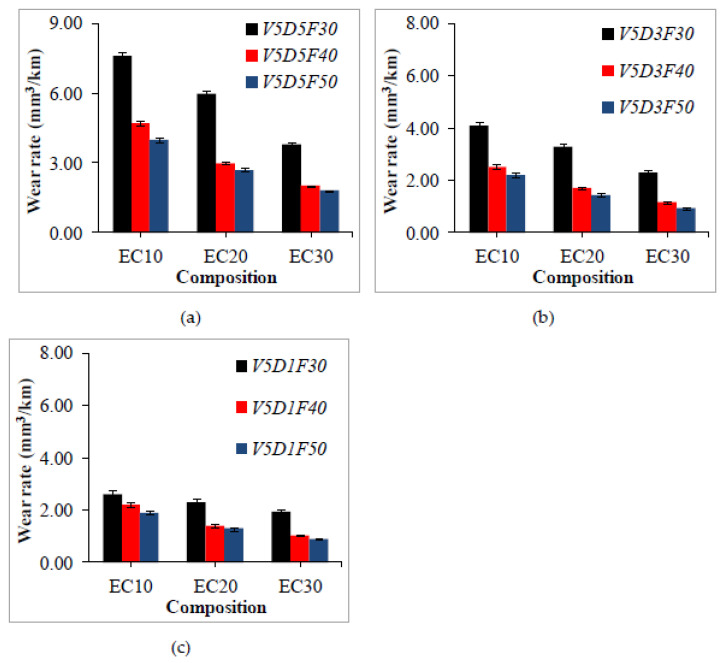
Experimentally measured wear rate at different sliding distances (**a**) 5 km (**b**) 3 km and (**c**) 1 km.

**Figure 6 polymers-13-02894-f006:**
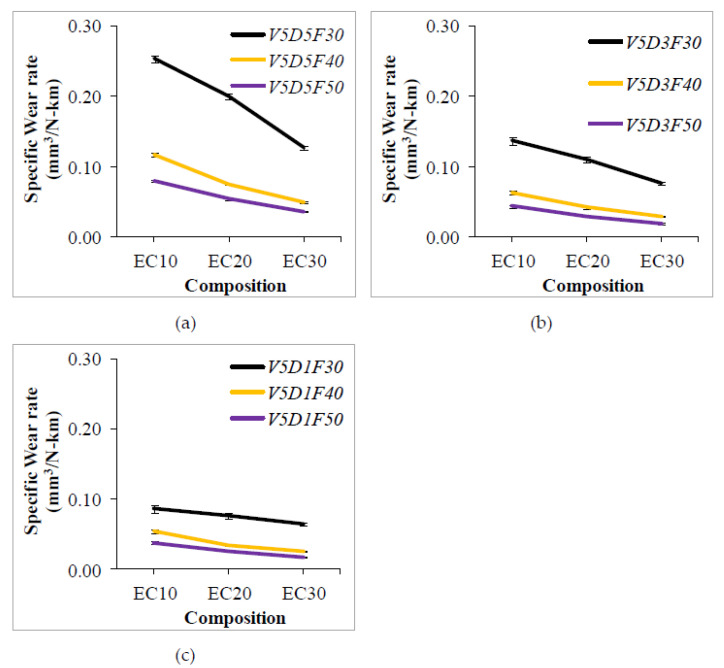
Experimentally measured specific wear rates at different sliding distances (**a**) 5 km (**b**) 3 km and (**c**) 1 km.

**Figure 7 polymers-13-02894-f007:**
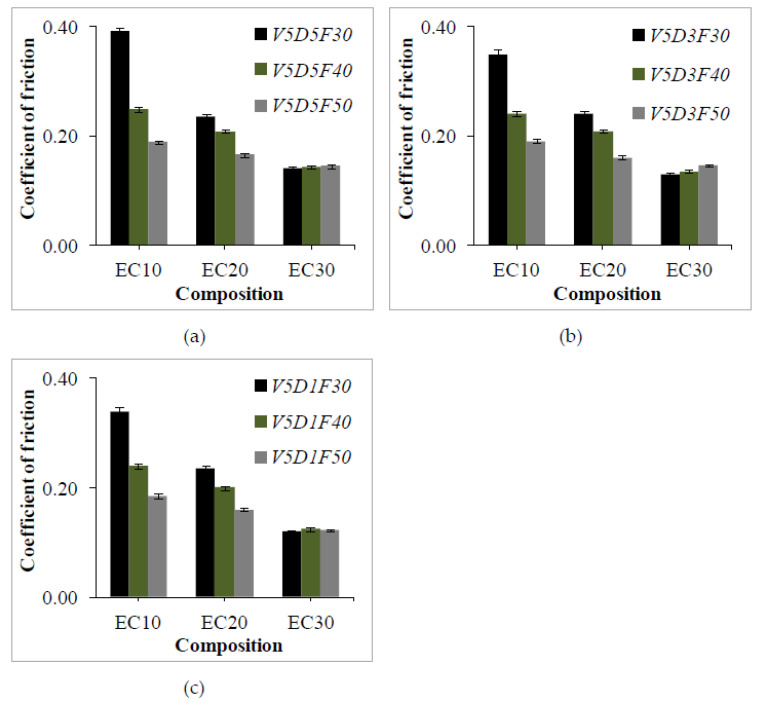
Experimentally measured coefficients of friction at different sliding distances (**a**) 5 km (**b**) 3 km and (**c**) 1 km.

**Figure 8 polymers-13-02894-f008:**
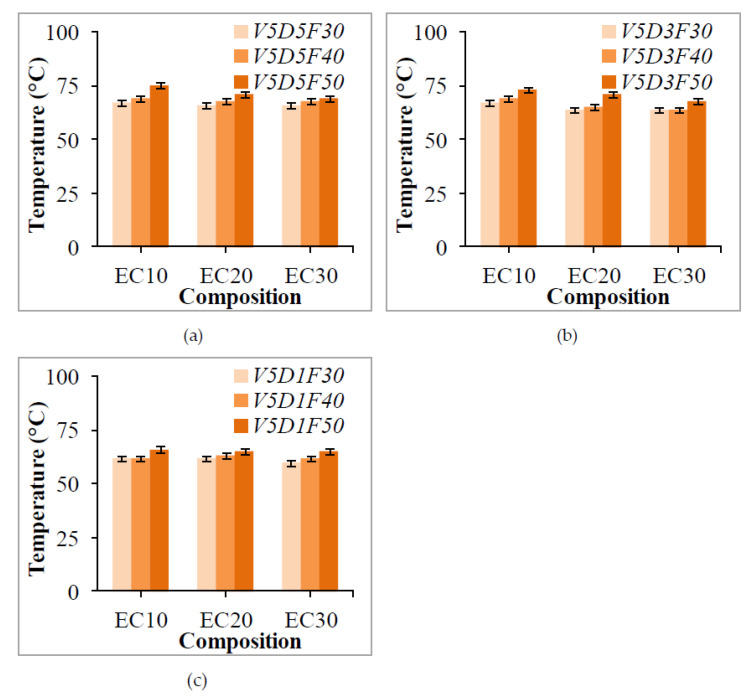
Disc temperature measured post-test at different sliding distances (**a**) 5 km (**b**) 3 km and (**c**) 1 km.

**Figure 9 polymers-13-02894-f009:**
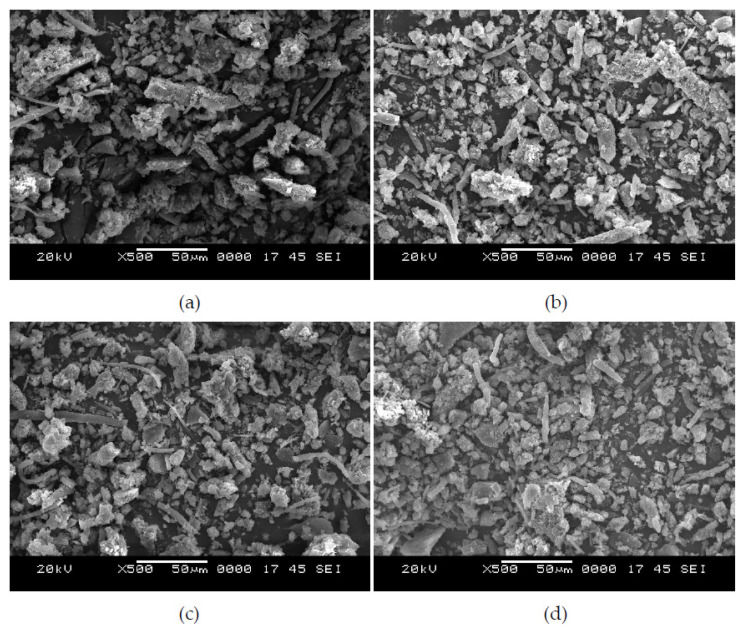
Morphology of wear debris of EC-10 at (**a**) 30 N and (**b**) 50 N; and EC-30 at (**c**) 30 N and (**d**) 50 N.

**Figure 10 polymers-13-02894-f010:**
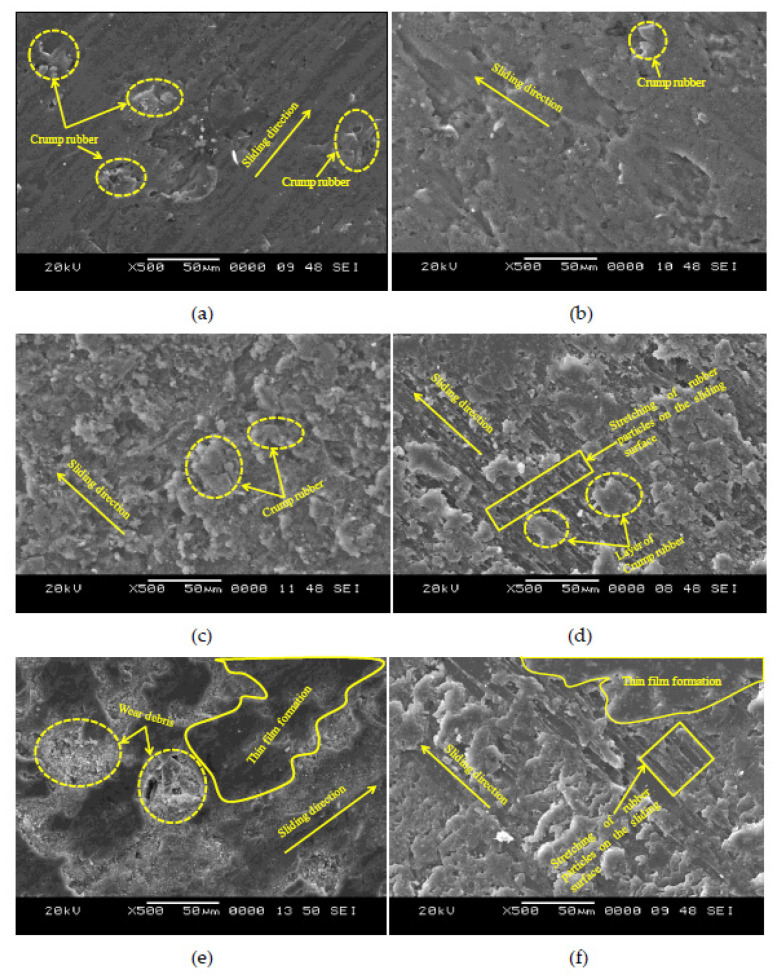
Scanning electron micrographs of EC-10 at (**a**) 30 and (**b**) 50 N; EC-20 at (**c**) 30 and (**d**) 50 N; and EC-30 at (**e**) 30 and (**f**) 50 N.

**Figure 11 polymers-13-02894-f011:**
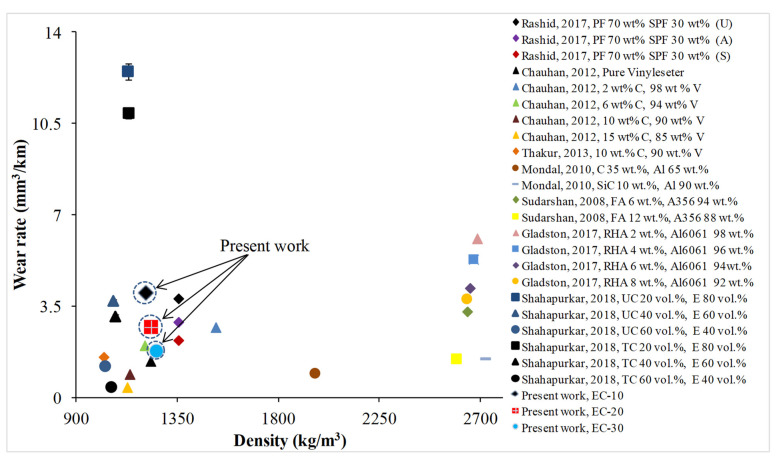
Plot of density against wear rate observed for different composites [12,47,48,49,50,51]. PF: phenolic resin; SPF: sugar palm fiber; U: untreated; A: alkali-treated; S: seawater-treated; C: cenosphere; V: vinyl ester; RHA: rice husk ash; FA: fly ash; SiC: silicon carbide; Al: aluminum; A356: aluminum–silicon alloy; Al6061: hardened aluminum alloy; UC: untreated cenosphere; TC: treated cenosphere.

**Table 1 polymers-13-02894-t001:** Crump rubber properties.

* Physical Properties	Unit
Density	1451 kg/m^3^
Young’s modulus	2600–2900 MPa
Tensile strength	40–70 MPa
Elongation at break	25–50%
Melting point	200 °C
Color	Black/blue

* As specified by the supplier.

**Table 2 polymers-13-02894-t002:** Levels of dry sliding wear parameters used in the study.

Parameters	Level 1	Level 2	Level 3
Filler content, (Vol.%)	10	20	30
Load, *F* (N)	30	40	50
Sliding distance, *D* (km)	1	3	5

**Table 3 polymers-13-02894-t003:** Experimental matrix and recorded response value.

Experiment No.	Crump Rubber (%)	Sliding Distance (km)	Normal Load (N)	Specific Wear Rate (mm^3^/N km)	COF
1	10	1	30	7.6	0.250
2	10	3	40	2	0.143
3	10	5	50	3.8	0.145
4	20	1	40	3	0.188
5	20	3	50	6	0.166
6	20	5	30	2.7	0.240
7	30	1	30	4	0.390
8	30	3	50	1.8	0.141
9	30	5	30	4.7	0.277

**Table 4 polymers-13-02894-t004:** S/N ratio, normalized S/N ratio, and grey coefficients.

S/N Ratio		Normalized S/N Ratio		Deviation Sequence		Grey Relational Coefficient	
(WR)	(COF)	(WR)	(COF)	(Δ_WR_)	(Δ_COF_)	(WR)	(COF)
−5.11	17.02	0.00	0.00	1.00	1.00	0.3333	0.3334
−6.02	16.89	0.07	0.01	0.93	0.99	0.3503	0.3365
−11.60	16.77	0.52	0.03	0.48	0.97	0.5094	0.3396
−9.54	14.52	0.35	0.28	0.65	0.72	0.4364	0.4103
−15.56	15.60	0.84	0.16	0.16	0.84	0.7525	0.3731
−8.63	12.40	0.28	0.52	0.72	0.48	0.4102	0.5100
−12.04	8.18	0.55	0.99	0.45	0.01	0.5286	0.9870
−17.62	12.04	1.00	0.56	0.00	0.44	0.9994	0.5316
−13.44	11.15	0.67	0.66	0.33	0.34	0.5995	0.5949

**Table 5 polymers-13-02894-t005:** Principal component values.

S.No.	Grey Relational Coefficients		Principal Component Value	
	Wear Rate	Coefficient of Friction	Wear Rate	Coefficient of Friction
1	0.3333	0.3334	0.167	0.167
2	0.3503	0.3365	0.175	0.168
3	0.5094	0.3396	0.255	0.170
4	0.4364	0.4103	0.218	0.205
5	0.7525	0.3731	0.376	0.186
6	0.4102	0.5100	0.205	0.255
7	0.5286	0.9870	0.264	0.493
8	0.9994	0.5316	0.500	0.266
9	0.5995	0.5949	0.300	0.297

**Table 6 polymers-13-02894-t006:** Eigenvalues of principal components.

Principal Component	Eigen Value	Percentage of Contribution	Cumulative
PC1	1.1855	59.27	59.27
PC2	0.8145	40.73	100

**Table 7 polymers-13-02894-t007:** Eigenvector value of the responses.

Responses	PC1	PC2
Specific wear rate	0.707	−0.707
Coefficient of friction	0.707	0.707

**Table 8 polymers-13-02894-t008:** Grey relational grades and their rankings.

S.No.	Grey Relational Grades		CGRG	Rank
	GRG_SWR_	GRG_COF_		
1	0.24	0.24	0.2357	9
2	0.25	0.24	0.2428	8
3	0.36	0.24	0.3001	6
4	0.31	0.29	0.2993	7
5	0.53	0.26	0.3979	4
6	0.29	0.36	0.3253	5
7	0.37	0.70	0.5357	2
8	0.71	0.38	0.5412	1
9	0.42	0.42	0.4222	3

**Table 9 polymers-13-02894-t009:** ANOVA analysis for the specific wear rate.

Source	DF	Adj SS	Adj MS	*F*-Value	*p*-Value	% of Contribution
Crump rubber (%)	2	0.11755	0.058774	10.18	0.089	52.60
Sliding distance (km)	2	0.01210	0.006051	1.05	0.488	5.40
Normal load (N)	2	0.09376	0.046878	8.12	0.110	42.00
Error	2	0.01155	0.005775			
Total	8	0.36673				

S = 0.0759959; R^2^ = 96.85%; R^2^ (adj) = 87.4%.

**Table 10 polymers-13-02894-t010:** ANOVA analysis for the coefficient of friction.

Source	DF	Adj SS	Adj MS	*F*-Value	*p*-Value	% of Contribution
Crump rubber (%)	2	0.147802	0.073901	1.77	0.360	86.76
Sliding distance (m)	2	0.014070	0.007035	0.17	0.855	8.24
Normal load (N)	2	0.008142	0.004071	0.10	0.911	5.00
Error	2	0.083294	0.041647			
Total	8	0.350820				

S = 0.0204076; R^2^ = 96.26%; R^2^ (adj) = 85.03%.

## Data Availability

Not applicable.

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
