# Peer review of "Leverage of Environmental Pollutant Crump Rubber on the Dry Sliding Wear Response of Epoxy Composites"

_polymers, 2021, doi:10.3390/polym13172894_

Round 1
Reviewer 1 Report
The idea of this paper seems interesting. However, there is a need for certain clarifications and additions:
- It would be worth clarifying where the results could be applied. Do the results have only cognitive or also applicable character?
- What were the dimensions of a flat samples and pins (diameters, length…) ?
- What was the hardness of dics samples?
- What was the roughness / surface topography parameters of the discs? Maybe isometric views of surface topography of selected composities can be added?
- On what basis did you choose L9 Taguchi plan? Much better choice is L27. I know that it reguires more experiments but in the case of friction and wear tests the larger number of experiments the better.
- What was the wear of pins in tribological tests? Did it not significantly affect the total wear of the system?
- Temperature plays a significant role in dry friction, especially when the friction coefficient is higher than 0.15-0.2. What was the temperature value at the different stages of the wear process? Were temperature changes correlated with e.g. an applied load or crump rubber content?
- The carelessness in the text should be corrected. For example, instead of references to figures and tables, the text says “Error! Reference source not found”; subsection 4.1 has no title; subsections 4.1.1, 4.2 and 4.3 start from picture without any word of introduction, etc.
Author Response
N/A

Reviewer 2 Report
Dear Editor: I would like to express my deep thanks for inviting me to review the manuscript ID: polymers-1293658
Title: Leverage of environmental pollutant crump rubber on the dry sliding wear response of epoxy composites
Authors: Kiran Shahapurkar, C. Venkatesh, Manzoore Elahi M. Soudagar, Irfan Anjum Badruddin, Pavan Shahapurkar, Ashraf Elfasakhany, MA Mujtaba, Md Irfanul Haque Siddiqui, Masood Ashraf Ali and Teuku Meurah Indra Mahlia
Comments:
Abstract
Please delete these sentences “Envisaged composites are capable of being used as tribo-materials in brake pad applications. Landfill burden and disposal issues of waste tires are addressed here by reinforcing varying amounts of crump rubber in the epoxy matrix”
Please rewrite according to you results
Introduction part:
Please discuss in aims and novelty of this work in Introduction section.
Materials and Methods:
Please check this sentence “Particle size analysis and micrograph of crump rubber are shown in Error! Reference source not found. Properties of crump rubber are depicted in Error! Reference source not found”. Please check throughout the manuscript.
Please include wear test number of each condition.
Results and discussion:
- Figures 3 and 4 are not clear, please provide clear images.
- Please include TG and DTA data that correlated with wear rate of different composition.
- There is no detail discussion throughout the manuscript.
Conclusion:
Please concise the conclusion part.
RECOMMENDATION
After reviewing the enclosed manuscript for “Polymers”, the present manuscript contains some kinds of scientific analysis but it is mandatory required to modify according to the preceding remarks. So, the manuscript can be accepted for publication after major revisions have been made.
Author Response
N/A

Round 2
Reviewer 1 Report
The authors provided mostly satisfactory answers and improvements. The paper can be published in this form.
Reviewer 2 Report
Authors addressed all of the comments in the revised manuscript.